# AIR-Net: Adaptive and Implicit Regularization Neural Network for Matrix Completion

## Abstract

Conventionally, the matrix completion (MC) model aims to recover a matrix from partially observed elements. Accurate recovery necessarily requires a regularization encoding priors of the unknown matrix/signal properly. However, encoding the priors accurately for the complex natural signal is difficult, and even then, the model might not generalize well outside the particular matrix type. This work combines adaptive and implicit low-rank regularization that captures the prior dynamically according to the current recovered matrix. Furthermore, we aim to answer the question: how does adaptive regularization affect implicit regularization? We utilize neural networks to represent Adaptive and Implicit Regularization and named the proposed model *AIR-Net*. Theoretical analyses show that the adaptive part of the AIR-Net enhances implicit regularization. In addition, the adaptive regularizer vanishes at the end, thus can avoid saturation issues. Numerical experiments for various data demonstrate the effectiveness of AIR-Net, especially when the locations of missing elements are not randomly chosen. With complete flexibility to select neural networks for matrix representation, AIR-Net can be extended to solve more general inverse problems.

## 1 Introduction

The matrix completion (MC) problem, which aims to recover a matrix $\boldsymbol{X}^* \in \mathbb{R}^{m \times n}$ from its partially observed elements, has arisen in numerous domains, ranging from computer vision (Wen et al., 2012), recommender system (Netflix, 2009), and drug-target interaction (DTI) (Mongia & Majumdar, 2020). This fundamental problem is ill-posed without assumptions on $\boldsymbol{X}^*$ since we have many completions. So it is essential to impose additional information or priors on the unknown matrix/signal.

To describe the prior for natural signal, or restrict the solution in the corresponding space is difficult. Classical methods for MC are mainly based on low-rank, sparsity or piece-wise smoothness assumption (Rudin et al., 1992; Buades et al., 2005; Romano et al., 2014; Dabov et al., 2007). These priors describe simple structural signal well, but may lead to a poor approximation of $\boldsymbol{X}^*$ with complex structures (Radhakrishnan et al., 2021) especially when the observed entries are not sampled uniformly at random. Recently, deep neural networks (DNN) have shown a strong ability in extracting complex structures from large datasets(Li et al., 2018; Mukherjee et al., 2021). However, such a large number of data sets cannot be obtained in many scenarios. Fortunately, DNN also works in solving some inverse problems without any extra training set (Ulyanov et al., 2018). Over-parametric DNN performs well on a single matrix is a mysterious phenomenon. One of the explanations is there exists **implicit regularization** during training (Arora et al., 2019; Xu et al., 2019; Rahaman et al., 2019; Chakrabarty & Maji, 2019). Although DNN with implicit regularization outperforms some classical methods, it is insufficient to describe the space of complex $\boldsymbol{X}^*$. Extra-explicit regularization can improve its performance in signal recovery (Metzler et al., 2018; Boyarski et al., 2019a; Liu et al., 2019; Li et al., 2020). However, such explicit priors are often valid only for specific data or sampling patterns. A more flexible regularization is required to meet practical MC problems.

We introduce flexibility in this paper by firstly representing the explicit regularization using DNN without any extra training set. The explicit regularization we begin with is Dirichlet Energy (DE), which is formulated as $\mathrm{tr}(\boldsymbol{X}^\top \boldsymbol{L} \boldsymbol{X})$, with $\boldsymbol{L}$ a Laplacian matrix describing the similarity between

columns. Note that $L$ in DE is fixed during iteration. Building an exact $L$ based on incomplete observation is very challenging. Therefore, we parameterize $L$ with DNN, and revise $L$ iteratively during training. Furthermore, We combine the learned DE, which is an **adaptive regularizer**, with implicit regularization to form a new regularization method for MC named AIR-Net. The interaction between explicit regularization and implicit regularization in solving MC problems is further studied. The results show that combining the two can obtain a new, more flexible regularization model and enhance the low-rank preference of implicit regularization. In many examples, AIR-NET has a more vital feature representation ability and more comprehensive application range and shows state-of-art performance.

## 2 ADAPTIVE AND IMPLICIT REGULARIZATION NEURAL NETWORK

Our model is proposed as follow:

$$\min_{\boldsymbol{X}, \boldsymbol{W}_i} \mathcal{L}_{all} = \mathcal{L}_{\mathbb{Y}} \left( \mathcal{A}\left(\boldsymbol{X}^*\right), \mathcal{A}(\boldsymbol{X})\right) + \sum_{i=1}^{N} \lambda_i \cdot \mathcal{R}_{\boldsymbol{W}_i}\left(\mathcal{T}_i\left(\boldsymbol{X}\right)\right) \tag{1}$$

where $\mathcal{A}\left(\boldsymbol{X}\right) = \boldsymbol{X}\mid_{\Omega} = \begin{cases} \boldsymbol{X}_{i,j}, & (i,j) \in \Omega \\ 0, & (i,j) \notin \Omega \end{cases}$ and $\Omega$ are the observed coordinates set, and the other entries are missing. Different from other regularization models for MC, here $\boldsymbol{X}$ is represented by a neural network which tends to be low-rank implicitly (Section 2.1), and $\mathcal{R}_{\boldsymbol{W}_i}$ is an adaptive regularization with a forward neural network represented Laplacian matrix(Section 2.2). The detailed notations will be introduced in the corresponding sections. A specific case of Equation 1 for matrix completion is given in Section 2.3.

### 2.1 DMF AS AN IMPLICIT REGULARIZATION

In order to make the model suitable for more matrix types, we need a more general data prior. The low-rank is a very general prior in various matrix types. There are two main ways to encode the low-rank prior into model: (a) Adding an explicit regularization term such as rank and nuclear norm (Candès & Recht, 2009; Lin et al., 2010). (b) Using a low-dimensional latent variable model to represent $\boldsymbol{X}$, including matrix factorization (MF) and its varieties (Koren et al., 2009; Fan & Cheng, 2018). The first case suffers from the saturation issue, which is induced by explicit regularization. The second one faces the problem of estimating a proper latent variable dimension.

Unlike the existing MF model, which constricts the size of the shared dimension of the factorized matrix, DMF can take a large shared dimension and still preserve the low-rank property without explicit regularization. This is the so-called implicit low-rank regularization of DMF:

$$\boldsymbol{X}(t) = \boldsymbol{W}^{[L-1]}(t)\boldsymbol{W}^{[L-2]}(t)\dots\boldsymbol{W}^{[1]}(t)\boldsymbol{W}^{[0]}(t) \in \mathbb{R}^{m \times n},$$

where $L$ is the depth of MF. $\boldsymbol{W}^{[l]}(t)$ represents the $l$-th matrix at the step $t$ during training. The results are given under a mild assumption 1 in Section A.2. This property helps avoiding dimension estimation and saturation issues. As for the details of the implicit low-rank we will discuss in Section A.2.

### 2.2 ADAPTIVE REGULARIZER

Apart from the low-rank prior, self-similarity is also a typical prior. The patch in the image and the rating behavior of users are all examples of self-similarity. For example, there is always a certain degree of self-similarity between the blocks in the image. A classical way to encode the similarity prior to $\boldsymbol{X}$ is Dirichlet Energy (DE) which is formulated as $\mathrm{tr}(\boldsymbol{X}^\top \boldsymbol{L}\boldsymbol{X})$. But DE will face two problems in applications: (a) $\boldsymbol{L}$ is unknown in MC problem, construct $\boldsymbol{L}$ based on incomplete $\boldsymbol{X}$ may induce worse prior. (b) The formulation of DE only encodes the similarity of the columns of $\boldsymbol{X}$. Other similarities such as block similarity cannot be captured. To address both of these issues, we parameterize $\boldsymbol{L}$ with DNN and replace $\boldsymbol{X}$ by a transformed $\mathcal{T}_i\left(\boldsymbol{X}\right)$ to capture the self-similarity flexibly.

The adaptive regularization is defined as

$$\mathcal{R}_{\boldsymbol{W}_i}\left(\mathcal{T}_i\left(\boldsymbol{X}\right)\right) = \mathrm{tr}\left(\mathcal{T}_i\left(\boldsymbol{X}\right)^\top \boldsymbol{L}_i\left(\boldsymbol{W}_i\right)\mathcal{T}_i\left(\boldsymbol{X}\right)\right), i = 1, 2, \dots, N$$

where $\boldsymbol{L}_i \in \mathbb{R}^{m_i \times m_i}$ is parameterized by $\boldsymbol{W}_i \in \mathbb{R}^{m_i \times m_i}$. To keep the Laplacian properties of $\boldsymbol{L}_i$, special design for the parameterized structure is important. We design a forward neural network which encodes the properties of Laplacian matrix in structure. The details are discussed in A.4. $\mathcal{T}_i : \mathbb{R}^{m \times n} \mapsto \mathbb{R}^{m_i \times n_i}$ transforms $\boldsymbol{X}$ into special domain, which makes the AIR-Net possible to capture various relationships embedded in data. The common choice can be $\mathcal{T}_i(\boldsymbol{X}) = \boldsymbol{X}$ which captures the relationship between columns. Regularization captures the relationship between rows when $\mathcal{T}_i(\boldsymbol{X}) = \boldsymbol{X}^\top$. Especially if $\mathcal{T}_i(\boldsymbol{X}) = \left[ \mathbf{vec}\left(\mathrm{block}(\boldsymbol{X})\right)_1, \mathbf{vec}\left(\mathrm{block}(\boldsymbol{X})\right)_2, \ldots, \mathbf{vec}\left(\mathrm{block}(\boldsymbol{X})\right)_{n_i} \right]$, where $\mathbf{vec}\left(\mathrm{block}(\boldsymbol{X})\right)_j \in \mathbb{R}^{m_i}$ is the vectorization of $j$-th block in $\boldsymbol{X}$ row by row, then the similarity among blocks can be obtained. A natural problem that arises is what the $\boldsymbol{L}_i$ looks like during training.

Obviously, $\mathcal{R}_{\boldsymbol{W}_i}$ reaches minimum when $\boldsymbol{L}_i = 0$, and this is called a trivial solution. The most exciting thing is that when we minimize Equation 1 with the gradient descent algorithm, $\{\boldsymbol{L}_i(\boldsymbol{W}_i(t))\}$ converges to a non-trivial solution. Another expected phenomenon is that $\mathcal{R}_{\boldsymbol{W}_i}$ vanishes at the end and will not cause the so-called saturation issue. The saturation issue is a bias term that dominates the overall estimation error due to explicit regularization. We illustrate these phenomena both by theoretical analysis (Theorem 2 in Section 3.2) and numerical experiments (Section 4).

## 2.3 AIR-NET FOR MC

In this subsection, we will focus our model on the MC problem. We select $\mathcal{T}_1(\boldsymbol{X}) = \boldsymbol{X}, \mathcal{T}_2(\boldsymbol{X}) = \boldsymbol{X}^\top, N = 2$ to capture the relationship both in rows and columns of $\boldsymbol{X}$. Overall, the theoretical analyses for general inverse problem is based on Equation 2.

$$\min_{\boldsymbol{X}, \boldsymbol{W}_r, \boldsymbol{W}_c} \mathcal{L}_{all} = \mathcal{L}_{\mathbb{Y}}\left(\mathcal{A}(\boldsymbol{X}^*), \mathcal{A}(\boldsymbol{X})\right) + \lambda_r \cdot \mathcal{R}_{\boldsymbol{W}_r}(\boldsymbol{X}) + \lambda_c \cdot \mathcal{R}_{\boldsymbol{W}_c}\left(\boldsymbol{X}^\top\right), \quad (2)$$

where $\boldsymbol{X} = \boldsymbol{W}^{[L-1]}\boldsymbol{W}^{[L-2]}\ldots\boldsymbol{W}^{[1]}\boldsymbol{W}^{[0]}$, $\mathcal{R}_{\boldsymbol{W}_r}(\boldsymbol{X}) = \boldsymbol{X}\boldsymbol{L}_r(\boldsymbol{W}_r)\boldsymbol{X}^\top$, $\mathcal{R}_{\boldsymbol{W}_c}\left(\boldsymbol{X}^\top\right) = \boldsymbol{X}^\top \boldsymbol{L}_c(\boldsymbol{W}_c)\boldsymbol{X}$. Specially, our experiments focus on the MC problem which can reform Equation 2 as follows:

$$\min_{\boldsymbol{W}^{[l]}, \boldsymbol{W}_r, \boldsymbol{W}_c} \mathcal{L}_{all} = \sum_{(i,j) \in \Omega} \left| \boldsymbol{X}_{ij} - \boldsymbol{X}_{ij}^* \right| + \lambda_r \cdot \mathcal{R}_{\boldsymbol{W}_r}(\boldsymbol{X}) + \lambda_c \cdot \mathcal{R}_{\boldsymbol{W}_c}\left(\boldsymbol{X}^\top\right), \quad (3)$$

with $l = 0, 1, \cdots, L-1$. The parameters in Equation 3 is updated by gradient descent algorithm or its variations. We stop the iteration until $\left|\mathcal{R}_{\boldsymbol{W}_r(T+1)} - \mathcal{R}_{\boldsymbol{W}_r(T)}\right| < \delta$ and $\left|\mathcal{R}_{\boldsymbol{W}_c(T+1)} - \mathcal{R}_{\boldsymbol{W}_c(T)}\right| < \delta$. The recovered matrix is $\hat{\boldsymbol{X}}(T) = \boldsymbol{W}^{[L-1]}(T) \cdots \boldsymbol{W}^{[0]}(T)$.

Some works which combine implicit and explicit regularization also can be regarded as a special case of Equation 1. Both the Total Variation (TV) and DE can be regarded as a fixed $\boldsymbol{L}$. Therefore, the framework of Equation 1 also contains DMF+TV (Li et al., 2020), DMF+DE (Boyarski et al., 2019a). So far, we cannot see any essential difference between Equation 3 and these models. We will illustrate the amazing properties of the model in the next section.

## 3 THEORETICAL ANALYSIS

In this section, we will analyze the properties based on the dynamics of Equation 3. Theorem 1 shows that our proposed regularization enhances the implicit low-rank regularization of DMF. Theorem 2 shows that the adaptive regularization will converge to a minimum while capturing the inner structure of data flexibly. Although this paper focus on MC problem, the following theoretical analyzes is satisfied for the general inverse problems. As $\mathcal{A}$ and $\mathcal{A}(\boldsymbol{X}^*)$ are fixed during optimization, we simplify $\mathcal{L}_{\mathbb{Y}}\left(\mathcal{A}(\boldsymbol{X}^*), \mathcal{A}(\boldsymbol{X})\right)$ as $\mathcal{L}_{\mathbb{Y}}(\boldsymbol{X})$ below. $\boldsymbol{U}_{i,j}$ is the $(i,j)$ th entry of $\boldsymbol{U}$, $\boldsymbol{U}_{:,k}$ and $\boldsymbol{U}_{k,:}$ are the $k$-th column and the $k$-th row of $\boldsymbol{U}$ respectively.

### 3.1 AIR-NET ENHANCES THE IMPLICIT LOW-RANK REGULARIZATION

To simplify the analysis, we keep $\boldsymbol{L}_r$ and $\boldsymbol{L}_c$ fixed. Then the $\mathcal{R}_{\boldsymbol{W}_i}, i = r, c$ only varies with $\boldsymbol{X}$. We will demonstrate what the adaptive regularizer brings to the implicit low-rank regularization.

**Theorem 1.** *Consider the following dynamics with initial data satisfying the balance initialization Assumption 1(see A.2):*

$$\dot{\boldsymbol{W}}^{[l]}(t) = -\frac{\partial}{\partial \boldsymbol{W}^{[l]}}\mathcal{L}_{all}(\boldsymbol{X}(t)), \quad t \geq 0, \quad l = 0, \ldots, L-1,$$

*where* $\mathcal{L}_{all}(\boldsymbol{X}) = \mathcal{L}_{\mathbb{Y}}(\boldsymbol{X}) + \lambda_r \cdot \mathcal{R}_{\boldsymbol{W}_r}(\boldsymbol{X}) + \lambda_c \cdot \mathcal{R}_{\boldsymbol{W}_c}(\boldsymbol{X})$. *Then for* $k = 1, 2, \ldots$, *we have*

$$
\begin{aligned}
\dot{\sigma}_k(t) = &- L\left(\sigma_k^2(t)\right)^{1-\frac{1}{L}}\left\langle\nabla_{\boldsymbol{W}}\mathcal{L}_{\mathbb{Y}}(\boldsymbol{X}(t)), \boldsymbol{U}_{:,k}(t)\boldsymbol{V}_{:,k}^{\top}(t)\right\rangle \\
&- 2L\left(\sigma_k^2(t)\right)^{\frac{3}{2}-\frac{1}{L}}\gamma_k(t),
\end{aligned}
\tag{4}
$$

*where* $\boldsymbol{X}(t) = \boldsymbol{U}(t)\boldsymbol{S}(t)\boldsymbol{V}^{\top}(t)$ *is the SVD for* $\boldsymbol{X}(t)$, $\boldsymbol{W} = \left[\boldsymbol{W}^{[0]}, \boldsymbol{W}^{[1]}, \ldots, \boldsymbol{W}^{[L-1]}\right]$, $\boldsymbol{X} = \sum_s \sigma_s \boldsymbol{U}_{:,s}\boldsymbol{V}_{:,s}^{\top}, \gamma_k(t) = \boldsymbol{U}_{:,k}^{\top}\boldsymbol{L}_r\boldsymbol{U}_{:,k} + \boldsymbol{V}_{:,k}^{\top}\boldsymbol{L}_c\boldsymbol{V}_{:,k} \geq 0$.

*Proof.* Directly calculate the gradient of $\mathcal{L}_{all}$ at $\boldsymbol{W}$ and utilize Equation 4 will obtain the result. The details of proof can be found in A.3. $\qquad\square$

Compared with the results of vanilla DMF whose order of $\sigma_k(t)$ is $2 - \frac{2}{L}$. This Theorem demonstrates that AIR-Net's $\sigma_k(t)$ has a higher dynamics order which is $3 - \frac{2}{L}$. Notice that the adaptive regularizer keeps $\gamma_k(t) \geq 0$. In this way, a bigger convergence speed gap appears between different singular values $\sigma_r$ than vanilla DMF. Therefore, the AIR-Net enhances the implicit tendency of DMF toward low-rank.

## 3.2 THE DYNAMICS OF ADAPTIVE REGULARIZER

Now suppose $\boldsymbol{X}$ is given and fixed. We focus on the converge property of $\mathcal{R}_{\mathcal{T}_i(\boldsymbol{W})}$ based on the evolutionary of the dynamics of $\boldsymbol{L}_i$. $\mathcal{R}_{\mathcal{T}_i(\boldsymbol{W})}$ vanishes at the end and avoids AIR-Net suffering from saturation issues.

**Theorem 2.** *Consider the gradient flow model, where* $\left\|\mathcal{T}_i(\boldsymbol{X})_{k,:}\right\|_F^2 = 1$ *and* $\mathcal{T}_i(\boldsymbol{X})_{k,l} > 0$. *If we initialize* $\boldsymbol{W}_i(0) = \varepsilon\boldsymbol{1}_{m_i \times m_i}$, *then* $\boldsymbol{W}(t)$ *will keep symmetric during optimization. We can get the following element-wise convergence relationship*

$$
\left|\boldsymbol{L}_{i(k,l)}(t) - \boldsymbol{L}_{i(k,l)}^*\right| \leq \begin{cases}
(m_i + 2k_i) \cdot \exp(-D \cdot t), & (k,l) \in \mathbb{C}_1 \\
\exp(-D \cdot t), & (k,l) \in \mathbb{C}_2 \\
(m_i - 1) \cdot (m_i + 2k_i)\exp(-D \cdot t), & k = l
\end{cases}
$$

*where* $\mathbb{C}_1 = \left\{(k,l) \mid k \neq l, \mathcal{T}_i(\boldsymbol{X})_{:,k} \neq \mathcal{T}_i(\boldsymbol{X})_{:,l}\right\}$, $\mathbb{C}_2 = \left\{(k,l) \mid k \neq l, \mathcal{T}_i(\boldsymbol{X})_{:,k} = \mathcal{T}_i(\boldsymbol{X})_{:,l}\right\}$,

$$
\boldsymbol{L}_{i(k,l)}^* = \begin{cases}
0, & (k,l) \in \mathbb{C}_1 \\
\gamma, & (k,l) \in \mathbb{C}_2 \\
-\sum_{l'=1,l'\neq l}^{m_i}\boldsymbol{L}_{i(k,l')}^*, & k = l
\end{cases}
$$

$\boldsymbol{L}_{i(k,l)}(t)$ *is* $(k,l)$-*th the element of* $\boldsymbol{L}_i(t)$, $\boldsymbol{1}_{m_i \times m_i}$ *is a matrix of all-one.* $\gamma = \frac{2}{\left|\{\boldsymbol{C}_{\tilde{k},l}=0\}\right|} = \frac{2}{m_i+2k_i}$, $D$ *is a constant defined in A.4 which equals to zero if and only if* $\boldsymbol{X} = \boldsymbol{1}_{m_i \times n_i}$.

*Proof.* We prove this theorem in A.4. $\qquad\square$

This Theorem gives the limit point $\boldsymbol{L}^*$ and convergence rate of $\boldsymbol{L}_i(t)$. $(k,l) \in \mathbb{C}_1$ $\boldsymbol{L}_i^*(k,l) = 0$. unless $\mathcal{T}_i(\boldsymbol{X})_{:,k} = \mathcal{T}_i(\boldsymbol{X})_{:,l}$ or $k = l$, that is to say, in the end, $\boldsymbol{L}_i^*$ will only think that the exact same columns in $\mathcal{T}_i(\boldsymbol{X})$ are related. $\boldsymbol{L}_{i(k,l)}(t)$ converges faster when $(k,l) \in \mathbb{C}_2$ than $(k,l) \in \mathbb{C}_1$. In another word, adaptive regularizer captures the similarity first. This convergence rate gap products a multi-scale similarity which will be discussed in Section 4.1. Additionally, it's not difficult to find $\mathcal{R}_{\boldsymbol{W}_i^*} = 0$, the convergence rate is given as follow:

**Corollary 1.** *In the setting of Theorem 2, we further have* $0 \leq \mathcal{R}_{\boldsymbol{W}_i}(t) \leq 2(m_i + 2k_i)(m_i - 1)m_i \cdot \exp(-Dt), i = 1, 2, \ldots, N$.

*Proof.* We prove this theorem in appendix A.5. □

According to Corollary 1, $\lim\limits_{t \to +\infty} \mathcal{R}_{\boldsymbol{W}_i}(t) \to 0$. Therefore, the regularization will vanish at the end and not induce the saturation issue.

**Remark 1.** *Notice that we have no restriction on specific $\mathcal{T}_i$, $\mathcal{A}$ or representation of $\boldsymbol{X}$ in the above proof. Therefore, the conclusion in this subsection is a general result for inverse problem.*

In this subsection, we demonstrate AIR-Net's fantastic theoretical properties. It can both enhance the implicit low-rank and avoid saturation issues. We will verify these properties and the effectiveness of AIR-Net in applications experimentally.

## 4 EXPERIMENTAL ANALYSIS

Now we demonstrate the adaptive properties of AIR-Net by numerical experiments: (a) $\boldsymbol{L}_r$ and $\boldsymbol{L}_c$ capture the structural similarity in data from large scale to small scale (Section 4.1); (b) The comprehensive similarity in all scales contribute to successful MC, therefore the adaptive regularizer is necessary (Section 4.2); (c) Because AIR-Net is **adaptive to data**, it avoids over-fitting and achieves good performance. (Section 4.3).

**Data type and sampling pattern** Three types of matrices are considered: gray-scale image, user-movie rating matrix, and drug-target interaction (DTI) data. Three standard test gray images of size $240 \times 240$(Monti et al., 2017) are included in the image type (Baboon, Barbara, and Cameraman). The user-movie rating matrix is Syn-Netflix which is of $150 \times 200$, and the DTI data has Ion channels (IC) and G protein-coupled receptor (GPCR) are shaped $210 \times 204$ and $223 \times 95$ respectively (Boyarski et al., 2019b; Mongia & Majumdar, 2020). The sampling patterns include random missing, patch missing and textural missing, which are listed in Figure 4. The random missing rate varies in different experiments, and the default is $30\%$.

**Parameter settings** We set $\lambda = \mu = \frac{\boldsymbol{X}^*_{\max} - \boldsymbol{X}^*_{\min}}{m_i \cdot n_i}$ to ensure the fidelity and the regularization are in the same order of magnitude, where $\boldsymbol{X}^*_{\max}$ and $\boldsymbol{X}^*_{\min}$ are maximum and minimum of $\boldsymbol{X}^*$. The $\delta$ is a threshold which we set as $\frac{mn}{1000}$ by default. All the parameters in AIR-Net are initialized with Gaussian distribution, which owns zero mean and $10^{-5}$ as its variance. The Adam is chosen as the optimization algorithm by default (Kingma & Ba, 2015).

### 4.1 AIR-NET CAPTURE RELATIONSHIP ADAPTIVE TO BOTH SPATIAL AND TIME DOMAIN

In this section, we will verify the previously proposed theorems. This section provides a few slices of $\boldsymbol{L}_r$ and $\boldsymbol{L}_c$ during training to demonstrate what AIR-Net can learn. The heatmap of $\boldsymbol{L}_r(t)$ and $\boldsymbol{L}_c(t)$ for Baboon at $t = 4000, 7000, 10000$ respectively are shown in Figure 1. The according results for Syn-Netflix are shown in Figure 5 in A.1. The first row shows the heatmap of $\boldsymbol{L}_r(t)$ and the second one shows the heatmap of $\boldsymbol{L}_c(t)$.

As Figure 1 shows, both $\boldsymbol{L}_r(t)$ and $\boldsymbol{L}_c(t)$ first appear many blocks ($t = 4000$). Specially, we sigh two of $\boldsymbol{L}_c(t = 4000)$ out. These blocks indicate that these corresponding blocks columns are highly related. These blocks correspond to columns in which the eyes of Baboon are located, which are indeed highly similar. However, the slight difference between these columns induces the relationship captured by adaptive regularizer focusing on the related columns ($t = 7000$), which is similar to TV(Rudin et al., 1992). The columns of Baboon are not fully the same. The regularization gradually vanishes ($t = 10000$), which matches the results of Theorem 2 (Figure 1). Except the gray-scale images, the results on Syn-Netflix give similar conclusion.

These results illustrate that AIR-Net captures the similarity from large scale to small scale. Meanwhile, a natural question is raised: does there exist a moment that both $\boldsymbol{L}_r$ and $\boldsymbol{L}_c$ are captured accurately? If yes, we can train AIR-Net with these fixed $\boldsymbol{L}_r$ and $\boldsymbol{L}_c$ to obtain better recovery performance. The experiments below show that the $\boldsymbol{L}_r$ and $\boldsymbol{L}_c$ captured by AIR-Net are necessary for MC.

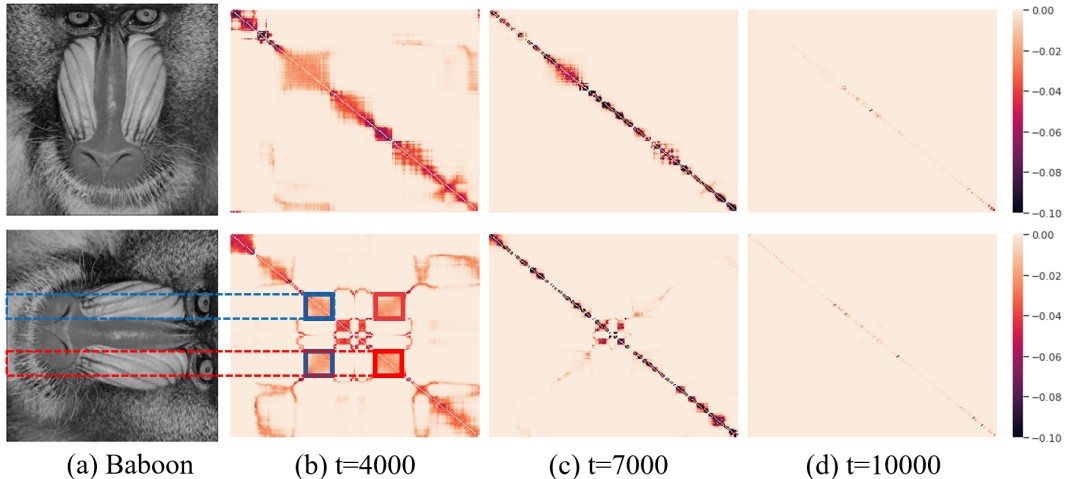

$$\begin{array}{cccc} \text{(a) Baboon} & \text{(b) t=4000} & \text{(c) t=7000} & \text{(d) t=10000} \end{array}$$

Figure 1: First row (column) shows the heatmap of $\boldsymbol{L}_r$ ($\boldsymbol{L}_c$) at different $t$. A darker color indicates a stronger similarity captured by the adaptive regularizer. The $(i, j)$-th element in the heatmap of $\boldsymbol{L}_r(t)$ has a darker color than the $(i, j')$-th element indicate that the $t$-th row is more related to $j$-th row compared with $j'$-th row. The area in the middle of the dotted line corresponding to the small block in the figure represents the part of the adaptive positive that is considered similar.

### 4.2 THE NECESSITY OF UTILIZING AN ADAPTIVE REGULARIZER

In this section, the necessity of adaptive updating $\boldsymbol{L}_r$ and $\boldsymbol{L}_c$ is explored. Let AIR-Net have a fixed regularizer, which is an adaptive regularizer learned at a specific step. The Normalized Mean Absolute Error (NMAE) is adopted to measure the distance between the recovered matrix $\hat{\boldsymbol{X}}$ and the actual matrix $\boldsymbol{X}^*$:

$$\text{NMAE} = \frac{1}{\left(\boldsymbol{X}_{\max}^* - \boldsymbol{X}_{\min}^*\right)|\bar{\Omega}|} \sum_{(i,j)\in\bar{\Omega}} \left|\hat{\boldsymbol{X}}_{ij} - \boldsymbol{X}_{ij}^*\right|,$$

where $\bar{\Omega}$ is the complement set of $\Omega$. We utilize the regularization captured by AIR-Net at $t = 4000, 7000, 9000$ respectively. All of the training hyper-parameters keep the same as AIR-Net. The Baboon under all the three missing patterns are tested.

Figure 2 shows how the NMAE changes with the epoch of training. AIR-Net, which updates the regularization during training, achieves the best performance in all missing patterns. The fixed regularization can accelerate the convergence speed of the algorithm. In random missing case, $\mathcal{R}_{\boldsymbol{W}_r}(9000)$ and $\mathcal{R}_{\boldsymbol{W}_c}(9000)$ is the best fixed regularizer among three time steps while other missing cases are $\mathcal{R}_{\boldsymbol{W}_r}(7000)$ and $\mathcal{R}_{\boldsymbol{W}_c}(7000)$. Fixed regularizer based methods will face two problems: (a) How to determine the best step? (b) How to estimate the regularization based on the partially observed matrix before training? These problems are not easy to solve. AIR-Net solves these problems from another perspective by updating the regularization during training. The adaptive property of AIR-Net is essential to the effectiveness of AIR-Net.

### 4.3 AIR-NET ADAPTIVE TO BOTH VARIES DATA AND MISSING PATTERN

Now we apply AIR-Net for matrix completion on three data types under different missing patterns.

**Peered methods** The peered methods include KNN(Goldberger et al., 2004), SVD(Troyanskaya et al., 2001), PNMC(Yang & Xu, 2020), DMF(Arora et al., 2019) and RDMF(Li et al., 2020) in image type. Here RDMF is replaced by DMF+DE(Boyarski et al., 2019a) because it is more suitable in the Syn-Netflix experiment.

**Avoid Over-fitting**. Figure 3 shows how the NMAE of DMF and AIR-Net changes with the training step. Compared with vanilla DMF, AIR-Net avoids over-fitting and achieves better performance on

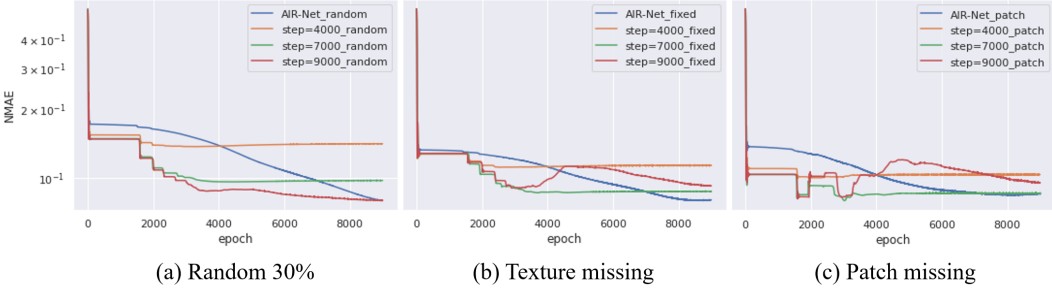

(a) Random 30%           (b) Texture missing           (c) Patch missing

Figure 2: Compare adaptive regularizer with fixed regularizer. The NMAE of recovered Baboon under three types of sampling, including random missing 30% pixels, patch missing, and texture missing, respectively. The blue line indicates the NMAE during training vanilla AIR-Net. Take the $L_r(t)$ and $L_c(t)$ out, $t$ equals 3000, 7000, 9000 respectively. The remind three lines in each figure indicate replacing the $L_r$ and $L_c$ with fixed $L_r(t)$ and $L_c(t)$.

all the three data types and missing patterns. The Syn-Netflix and DIT data can be found in Figure 6 at A.1.

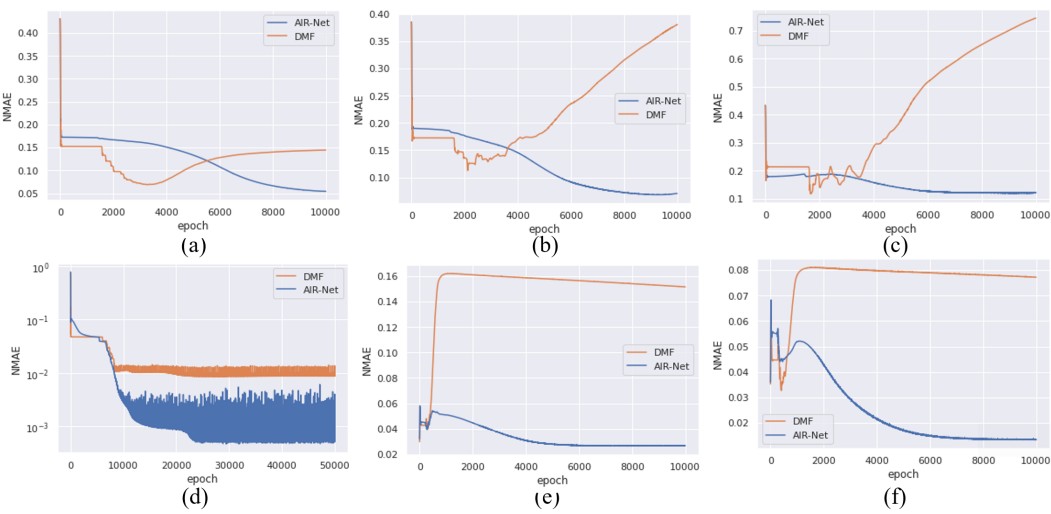

Figure 3: NMAE during training of DMF and AIR-Net. All the figures show the NMAE changes with the training step. The first row shows the results of Cameraman with (a) random missing (The proportion of different percentage figures show that the random missing) (b) textural missing (c) patch missing. The second row shows the remind data type with random missing respectively, including (d) Syn-Netflix, (e)IC and (f) GPCR.

**Adaptive to data**. Our proposed method achieves the best-recovered performance in most tasks. Table 1 shows the efficacy of AIR-Net on the various data types. More surprising is that our methods perform better than other methods, which are well designed for the particular data type. The recovered results are shown in Figure 4. In this figure, the existing methods perform well on specific missing pattern data. Such as the RDMF achieved good performance on the random missing case but performed not OK on reminding missing patterns. PNMC completed the patch missing well while obtaining worse results on texture missing. Thanks to the proposed model's adaptive properties, our method achieves promising results both visually and by numerical measures.

Table 1: NMAE values of compared algorithms with different missing patterns in different images. The bold font-type indicates the best performance. KNN(Goldberger et al., 2004), SVD(Troyanskaya et al., 2001), PNMC(Yang & Xu, 2020), DMF(Arora et al., 2019), DMF+DE(Boyarski et al., 2019a), RDMF(Li et al., 2020), AIR-Net(proposed). Some elements without value are not suitable for that data type.

| Data | Missing | KNN | SVD | PNMC | DMF | RDMF | DMF+DE | Proposed |
|---|---|---|---|---|---|---|---|---|
| | 30% | 0.083 | 0.0621 | 0.0622 | 0.0613 | 0.0494 | - | **0.0471** |
| Barbara | Patch | 0.1563 | 0.2324 | 0.2055 | 0.7664 | 0.3025 | - | **0.1195** |
| | Texture | 0.0712 | 0.1331 | 0.1100 | 0.3885 | 0.1864 | - | **0.0692** |
| | 30% | 0.0831 | 0.1631 | 0.0965 | 0.2134 | 0.0926 | - | **0.0814** |
| Baboon | Patch | 0.1195 | 0.1571 | 0.1722 | 0.8133 | 0.2111 | - | **0.1316** |
| | Texture | 0.1237 | 0.1815 | 0.1488 | 0.5835 | 0.2818 | - | **0.1208** |
| | 70% | 0.0032 | 0.0376 | - | 0.0003 | - | 0.0008 | **0.0002** |
| Syn-Netflix | 75% | 0.0046 | 0.0378 | - | 0.0004 | - | 0.0009 | **0.0003** |
| | 80% | 0.0092 | 0.0414 | - | 0.0014 | - | 0.0012 | **0.0007** |
| IC | 20% | 0.0169 | 0.0547 | - | 0.0773 | - | 0.0151 | **0.0134** |
| GPCR | 20% | 0.0409 | 0.0565 | - | 0.1513 | - | **0.0245** | 0.0271 |

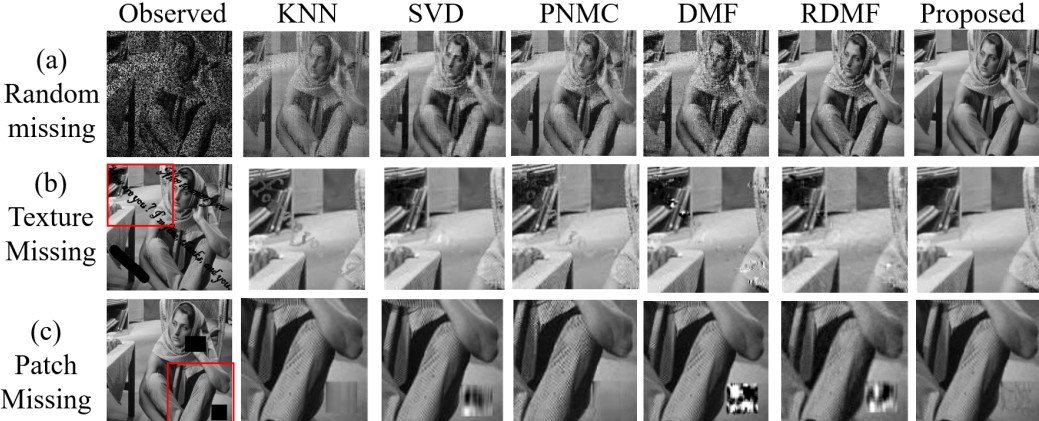

Figure 4: Compared KNN(Goldberger et al., 2004), SVD(Troyanskaya et al., 2001), PNMC(Yang & Xu, 2020), DMF(Arora et al., 2019), RDMF(Li et al., 2020), AIR-Net(proposed) on Babara with three types of data respectively.

## 5 CONCLUSION

We have proposed AIR-Net which aims to solve the MC problem without knowning the prior in advance. We show that our AIR-Net can adaptively learn the regularization according to different data at different training steps. In addition, we demonstrate that AIR-Net can avoid the saturation issue and over-fitting issue simultaneously. In fact, the AIR-Net is a general framework for solving the inverse problem. In the future work, we will combine other implicit regularization such as F-Principle(Xu et al., 2019) with more flexible $\mathcal{T}_i$ for other inverse problems.

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

## A  APPENDIX

### A.1  EXPERIMENTS RESULTS

In this section, we place the experiments mentioned before. Figure 5 shows the heatmap of $\boldsymbol{L}_r$ and $\boldsymbol{L}_c$ learned by adaptive regularizer. Eventually, adaptive regularizer obtain the $\boldsymbol{L}_r$ and $\boldsymbol{L}_c$ which are highly similar to real $\hat{\boldsymbol{L}}_r$ and $\hat{\boldsymbol{L}}_c$ in first column.

Figure 6 shows the NMAE of Syn-Netflix, IC and GPCR during training, respectively. This experiment result also shows the ability to avoid over-fitting.

### A.2  INTRODUCTION OF DMF

**Assumption 1.** *Factor matrices are balanced at initialization, i.e.,*

$$\boldsymbol{W}^{[l+1]^\top}(0)\boldsymbol{W}^{[l+1]}(0) = \boldsymbol{W}^{[l]}(0)\boldsymbol{W}^{[l]^\top}(0), \quad l = 0, \ldots, L-2.$$

Under this assumption, Arora et al. studied the gradient flow of the non-regularized risk function $\mathcal{L}_{\mathbb{Y}}$, i.e.,

$$\dot{\boldsymbol{W}}^{[l]}(t) = -\frac{\partial}{\partial \boldsymbol{W}^{[l]}} \mathcal{L}_{\mathbb{Y}}\left(\boldsymbol{X}(t)\right), \quad t \geq 0, \quad l = 0, \ldots, L-1, \tag{5}$$

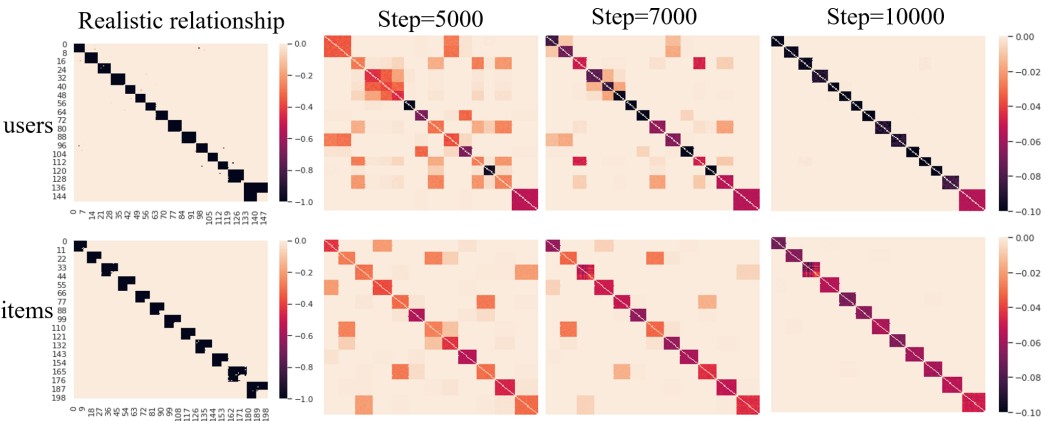

Figure 5: The first column shows the realistic relationship among columns and rows respectively. The remind three columns are the Laplacian matrix learned by AIR at different step.

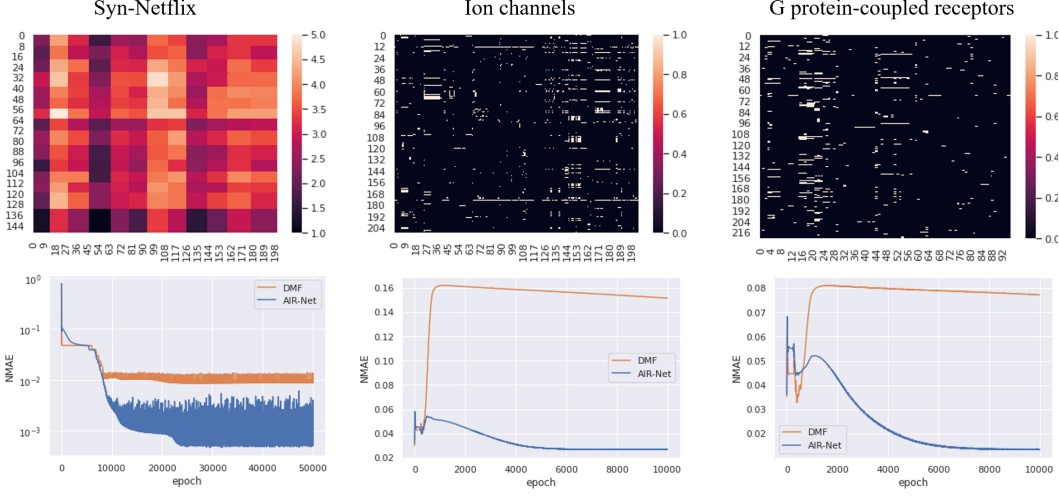

Figure 6: The first row shows the real data of Syn-Netflix, IC and GPCR respectively. The second row shows the corresponding NMAE during training.

where the empirical risk $\mathcal{L}_{\mathbb{Y}}$ can be any analytic function of $\boldsymbol{X}(t)$. According to the analyticity of $\mathcal{L}_{\mathbb{Y}}$, $\boldsymbol{X}(t)$ has the following singular value decomposition where each matrix is an analytic function of $t$:

$$\boldsymbol{X}(t) = \boldsymbol{U}(t)\boldsymbol{S}(t)\boldsymbol{V}^\top(t),$$

where $\boldsymbol{U}(t) \in \mathbb{R}^{m,\min\{m,n\}}$, $\boldsymbol{S}(t) \in \mathbb{R}^{\min\{m,n\},\min\{m,n\}}$, and $\boldsymbol{V}(t) \in \mathbb{R}^{\min\{m,n\},n}$ are analytic functions of $t$; and for every $t$, the matrices $\boldsymbol{U}(t)$ and $\boldsymbol{V}(t)$ have orthonormal columns, while $\boldsymbol{S}(t)$ is diagonal (its diagonal entries may be negative and may appear in any order). The diagonal entries of $\boldsymbol{S}(t)$, which we denote by $\sigma_1(t), \ldots, \sigma_{\min\{m,n\}}(t)$, are signed singular values of $\boldsymbol{X}(t)$. The columns of $\boldsymbol{U}(t)$ and $\boldsymbol{V}(t)$, denoted by $\boldsymbol{U}_1(t), \ldots, \boldsymbol{U}_{\min\{m,n\}}(t)$ and $\boldsymbol{V}_1(t), \ldots, \boldsymbol{V}_{\min\{m,n\}}(t)$, are the corresponding left and right singular vectors respectively. Based on these notation, Arora derive the following singular values evolutionary dynamics equation.

**Proposition 1** ((Arora et al., 2019, Theorem 3)). *Consider the dynamics Equation 5 with initial data satisfying Assumption 1. Then the signed singular values $\sigma_k(t)$ of the product matrix $\boldsymbol{X}(t)$ evolve by:*

$$\dot{\sigma}_k(t) = -L\left(\sigma_k^2(t)\right)^{1-\frac{1}{L}} \left\langle \nabla_{\boldsymbol{X}}\mathcal{L}_{\mathbb{Y}}(\boldsymbol{X}(t)), \boldsymbol{U}_{:,k}(t)\boldsymbol{V}_{:,k}^\top(t)\right\rangle,$$
$$k = 1, \ldots, \min\{m, n\}. \tag{6}$$

If the matrix factorization is non-degenerate, i.e., has depth $L \geq 2$, the singular values need not be signed (we may assume $\sigma_k(t) \geq 0$ for all $t$).

Arora et al. claimed the terms $\left(\sigma_k^2(t)\right)^{1-\frac{1}{L}}$ enhance the movement of large singular values, and on the other hand, attenuate that of small ones. The enhancement/attenuation becomes more significant as $L$ grows.

### A.3 PROOF OF THEOREM 1

We first give the details of the proposed adaptive regularizer with a iterative definition:

$$\begin{cases} \mathcal{R}_{\boldsymbol{W}_i}\left(\mathcal{T}_i\left(\boldsymbol{X}\right)\right) = \operatorname{tr}\left(\mathcal{T}_i\left(\boldsymbol{X}\right)^\top \boldsymbol{L}_i \mathcal{T}_i\left(\boldsymbol{X}\right)\right) \\ \boldsymbol{L}_i = \left(\boldsymbol{A}_i^* \cdot \mathbf{1}_{m_i \times m_i}\right) \odot \boldsymbol{I}_{m_i} - \boldsymbol{A}_i^* \\ \boldsymbol{A}_i^* = \boldsymbol{A}_i \odot \left(\mathbf{1}_{m_i \times m_i} - \boldsymbol{I}_{m_i}\right) \\ \boldsymbol{A}_i = \dfrac{exp(\boldsymbol{W}_i + \boldsymbol{W}_i^\top)}{\|\exp(\boldsymbol{W}_i)\|_1} \end{cases},$$

**Theorem 1.** *Consider the following dynamics with initial parameters satisfying Assumption 1:*

$$\dot{\boldsymbol{W}}^{[l]}(t) = -\frac{\partial}{\partial \boldsymbol{W}^{[l]}}\mathcal{L}_{all}(\boldsymbol{X}(t)), \quad t \geq 0, \quad l = 0, \ldots, L-1,$$

*where $\mathcal{L}_{all}\left(\boldsymbol{X}\right) = \mathcal{L}_{\mathbb{Y}}(\boldsymbol{X}) + \lambda_r \cdot \mathcal{R}_{\boldsymbol{W}_r}\left(\boldsymbol{X}\right) + \lambda_c \cdot \mathcal{R}_{\boldsymbol{W}_c}\left(\boldsymbol{X}\right)$. Then we have for any $k = 1, 2, \ldots$*

$$\dot{\sigma}_k(t) = -L\left(\sigma_k^2(t)\right)^{1-\frac{1}{L}} \left\langle \nabla_{\boldsymbol{W}}\mathcal{L}_{\mathbb{Y}}(\boldsymbol{X}(t)), \boldsymbol{U}_{:,k}(t)\boldsymbol{V}_{:,k}^\top(t)\right\rangle$$
$$- 2L\left(\sigma_k^2(t)\right)^{\frac{3}{2}-\frac{1}{L}} \gamma_k(t),$$

*where $\boldsymbol{X}(t) = \boldsymbol{U}(t)\boldsymbol{S}(t)\boldsymbol{V}^\top(t)$, $\boldsymbol{X} = \sum_s \sigma_s \boldsymbol{U}_{:,s}\boldsymbol{V}_{:,s}^\top, \gamma_k(t) = \boldsymbol{U}_{:,k}^\top \boldsymbol{L}_r \boldsymbol{U}_{:,k} + \boldsymbol{V}_{:,k}^\top \boldsymbol{L}_c \boldsymbol{V}_{:,k} \geq 0$.*

*Proof.* This is proved by direct calculation:

$$\nabla_{\boldsymbol{W}}\left(\lambda_r \cdot \mathcal{R}_r + \lambda_c \cdot \mathcal{R}_c\right) = \frac{\partial \operatorname{tr}\left(\lambda_r \cdot \boldsymbol{X}^\top \boldsymbol{L}_r \boldsymbol{X} + \lambda_c \cdot \boldsymbol{X}\boldsymbol{L}_c\boldsymbol{X}^\top\right)}{\partial \boldsymbol{X}}$$
$$= 2\lambda_r \cdot \boldsymbol{L}_r \boldsymbol{X} + 2\lambda_c \cdot \boldsymbol{X}\boldsymbol{L}_c$$
$$= 2\lambda_r \cdot \boldsymbol{L}_r \sum_s \sigma_s \boldsymbol{U}_{:,s}\boldsymbol{V}_{:,s}^\top + 2\lambda_c \cdot \sum_s \sigma_s \boldsymbol{U}_{:,s}\boldsymbol{V}_{:,s}^\top \boldsymbol{L}_c.$$

Note that

$$\langle \boldsymbol{V}_{:,s}, \boldsymbol{V}_{s'}\rangle = \langle \boldsymbol{U}_{:,s}, \boldsymbol{U}_{s'}\rangle = \delta_{ss'} = \begin{cases} 1, & s = s', \\ 0. & s \neq s'. \end{cases}$$

Therefore

$$
\begin{aligned}
\boldsymbol{U}_{:,k}^{\top}\left(\nabla_{\boldsymbol{W}}\left(\lambda_r \cdot \mathcal{R}_r + \lambda_c \cdot \mathcal{R}_c\right)\right) \boldsymbol{V}_{:,k} &= 2\sigma_k(\lambda_r \cdot \boldsymbol{U}_{:,k}^{\top} \boldsymbol{L}_r \boldsymbol{U}_{:,k} + \lambda_c \cdot \boldsymbol{V}_{:,k}^{\top} \boldsymbol{L}_c \boldsymbol{V}_{:,k}) \\
&= 2\sigma_k \gamma_k(t),
\end{aligned}
$$

where the term $\gamma_k(t) = 2\sigma_k(\lambda_r \cdot \boldsymbol{U}_{:,k}^{\top} \boldsymbol{L}_r \boldsymbol{U}_{:,k} + \lambda_c \cdot \boldsymbol{V}_{:,k}^{\top} \boldsymbol{L}_c \boldsymbol{V}_{:,k}) \geq 0$. Furthermore, according to Equation 6, we have $\boldsymbol{U}_{:,k}^{\top} \nabla_{\boldsymbol{W}} \mathcal{L}_{\mathbb{Y}} \boldsymbol{V}_{:,k} = -L\left(\sigma_k^2(t)\right)^{1-\frac{1}{L}}\left\langle \nabla_{\boldsymbol{W}} \mathcal{L}_{\mathbb{Y}}(\boldsymbol{X}(t)), \boldsymbol{U}_{:,k}(t) \boldsymbol{V}_{:,k}^{\top}(t) \right\rangle$.

Finally, according to $\dot{\boldsymbol{W}}^{[l]}(t) = -\frac{\partial}{\partial \boldsymbol{W}^{[l]}} \mathcal{L}_{all}(\boldsymbol{X}(t))$, we have

$$
\dot{\sigma}_k(t) = -L\left(\sigma_k^2(t)\right)^{1-\frac{1}{L}}\left\langle \nabla_{\boldsymbol{W}} \mathcal{L}_{\mathbb{Y}}(\boldsymbol{X}(t)), \boldsymbol{U}_{:,k}(t) \boldsymbol{V}_{:,k}^{\top}(t)\right\rangle - 2L\left(\sigma_k^2(t)\right)^{\frac{3}{2}-\frac{1}{L}} \gamma_k(t).
$$

□

### A.4 PROOF OF THEOREM 2

**Proposition 2.** $\nabla_{\boldsymbol{W}_i}\left(\mathcal{R}_{\boldsymbol{W}_i}(\boldsymbol{X})\right) = 2\boldsymbol{C} \odot \boldsymbol{A}_i - 2\mathrm{tr}\left(\boldsymbol{C} \boldsymbol{A}_i'\right) \boldsymbol{A}_i'$, where $\boldsymbol{A}_i' = \frac{\exp(\boldsymbol{W}_i)}{\|\exp(\boldsymbol{W}_i)\|_1}$, $\boldsymbol{A}_i = \boldsymbol{A}_i' + \boldsymbol{A}_i'^{\top}$ and $\boldsymbol{C} = \mathbf{1}_{m_i \times m_i} \cdot \left(\mathcal{T}_i(\boldsymbol{X}) \mathcal{T}_i(\boldsymbol{X})^{\top} \odot \boldsymbol{I}_{m_i}\right) - \mathcal{T}_i(\boldsymbol{X}) \mathcal{T}_i(\boldsymbol{X})^{\top}$.

*Proof.* We denote $\boldsymbol{X} = \mathcal{T}_i(\boldsymbol{X}) \in \mathbb{R}^{m_i \times n_i}$, then we consider $d\left[\mathrm{tr}\left(\boldsymbol{X}^{\top} \boldsymbol{L}_i \boldsymbol{X}\right)\right]$

$$
\begin{aligned}
&d\left[\mathrm{tr}\left(\boldsymbol{X}^{\top} \boldsymbol{L}_i \boldsymbol{X}\right)\right] \\
=&\mathrm{tr}\left[d\left(\boldsymbol{L}_i \boldsymbol{X} \boldsymbol{X}^{\top}\right)\right] \\
=&\mathrm{tr}\left[(d\boldsymbol{A}_i \odot (\mathbf{1}_{m_i \times m_i} - \boldsymbol{I}_{m_i}) \cdot \mathbf{1}_{m_i \times m_i}) \odot \boldsymbol{I}_n \cdot \boldsymbol{X} \boldsymbol{X}^{\top}\right. \\
&\left. - d\boldsymbol{A}_i \odot (\mathbf{1}_{m_i \times m_i} - \boldsymbol{I}_{m_i}) \boldsymbol{X} \boldsymbol{X}^{\top}\right] \\
=&\mathrm{tr}\left[\left(\boldsymbol{X} \boldsymbol{X}^{\top}\right)^{\top}\left(\boldsymbol{I}_{m_i} \odot (d\boldsymbol{A}_i \odot (\mathbf{1}_{m_i \times m_i} - \boldsymbol{I}_{m_i}) \cdot \mathbf{1}_{m_i \times m_i})\right)\right. \\
&\left. - \left(\boldsymbol{X} \boldsymbol{X}^{\top}\right)^{\top}\left((\mathbf{1}_{m_i \times m_i} - \boldsymbol{I}_{m_i}) \odot d\boldsymbol{A}_i\right)\right] \\
=&\mathrm{tr}\left[\left(\boldsymbol{X} \boldsymbol{X}^{\top} \odot \boldsymbol{I}_{m_i}\right)^{\top} d\boldsymbol{A}_i \odot (\mathbf{1}_{m_i \times m_i} - \boldsymbol{I}_{m_i}) \cdot \mathbf{1}_{m_i \times m_i}\right. \\
&\left. - \left(\boldsymbol{X} \boldsymbol{X}^{\top} \odot (\mathbf{1}_{m_i \times m_i} - \boldsymbol{I}_{m_i})\right)^{\top} d\boldsymbol{A}_i\right] \\
=&\mathrm{tr}\left[\left(\left(\boldsymbol{X} \boldsymbol{X}^{\top} \odot \boldsymbol{I}_{m_i}\right) \mathbf{1}_{m_i \times m_i}\right)^{\top}\left(d\boldsymbol{A}_i \odot (\mathbf{1}_{m_i \times m_i} - \boldsymbol{I}_{m_i})\right)\right. \\
&\left. - \left(\boldsymbol{X} \boldsymbol{X}^{\top} \odot (\mathbf{1}_{m_i \times m_i} - \boldsymbol{I}_{m_i})\right)^{\top} d\boldsymbol{A}_i\right] \\
=&\mathrm{tr}\left[\left(\left(\left(\boldsymbol{X} \boldsymbol{X}^{\top} \odot \boldsymbol{I}_{m_i}\right) \cdot \mathbf{1}_{m_i \times m_i}\right) \odot (\mathbf{1}_{m_i \times m_i} - \boldsymbol{I}_{m_i}) - \left(\boldsymbol{X} \boldsymbol{X}^{\top} \odot (\mathbf{1}_{m_i \times m_i} - \boldsymbol{I}_{m_i})\right)^{\top}\right) d\boldsymbol{A}_i\right] \\
=&\mathrm{tr}\left[\left(\left(\boldsymbol{X} \boldsymbol{X}^{\top} \odot \boldsymbol{I}_{m_i}\right) \mathbf{1}_{m_i \times m_i} - \boldsymbol{X} \boldsymbol{X}^{\top}\right) d\boldsymbol{A}_i\right]
\end{aligned}
$$

We denote $C = \left(XX^\top \odot I_{m_i}\right) \mathbf{1}_{m_i \times m_i} - XX^\top$, $S_{W_i} = \mathbf{1}_{m_i}^\top \cdot exp\left(W_i\right) \cdot \mathbf{1}_{m_i}$, then

$$
\begin{aligned}
&d\left[\text{tr}\left(X^\top L_i X\right)\right]\\
&=\text{tr}\left(C d A_i\right)\\
&=\frac{1}{S_{W_i}^2}\text{tr}\left[C\left(S_{W_i}\cdot\exp(W_i+W_i^\top)\odot d(W_i+W_i^\top)\right)\right.\\
&\quad\left.-C\left(\mathbf{1}_{m_i}^\top\left(\exp(W_i)\odot dW_i\right)\mathbf{1}_{m_i}\right)\exp(W_i+W_i^\top)\right]\\
&=\text{tr}\left[C\cdot\left(A_i\odot d\left(W_i+W_i^\top\right)\right)\right]\\
&\quad-\frac{1}{S_{W_i}^2}\text{tr}\left[\mathbf{1}_{m_i\times m_i}\left(\exp(W_i)\odot dW_i\right)\right]\cdot\text{tr}\left[C\cdot exp\left(W_i+W_i^\top\right)\right]\\
&=\text{tr}\left[C\cdot\left(A_i\odot d\left(W_i+W_i^\top\right)\right)\right]\\
&\quad-\frac{1}{S_{W_i}^2}\text{tr}\left[\left(\mathbf{1}_{m_i\times m_i}\odot\exp(W_i)\right)dW_i\right]\cdot\text{tr}\left[C\cdot exp\left(W_i+W_i^\top\right)\right]\\
&=\text{tr}\left[C\cdot\left(A_i\odot d\left(W_i+W_i^\top\right)\right)\right]-\text{tr}\left[\text{tr}\left(C\cdot A_i\right)\cdot A_i' dW_i\right]\\
&=\text{tr}\left[\left(\left(C^\top\odot A_i\right)^\top+C^\top\odot A_i-\text{tr}\left(C\cdot A_i\right)A_i'\right)dW_i\right]
\end{aligned}
$$

Therefore,

$$
\begin{aligned}
\nabla_{W_i}\text{tr}\left(X^\top L_i X\right)&=\left(C^\top\odot A_i\right)^\top+C^\top\odot A_i-\text{tr}\left(C\cdot A_i\right)A_i'\\
&=2C\odot A_i-2\text{tr}\left(CA_i'\right)A_i'
\end{aligned}
$$

Notice that $X = \mathcal{T}_i\left(X\right) \in \mathbb{R}^{m_i \times n_i}$, the proposition is proved. $\qquad\square$

**Theorem 2.** *Consider the gradient flow model, assume $\left\|\mathcal{T}_i\left(X\right)_{k,:}\right\|_F^2 = 1$ and $\mathcal{T}_i\left(X\right)_{k,l} > 0$, if we initialize $W_i\left(0\right) = \varepsilon\mathbf{1}_{m_i\times m_i}$, then $W\left(t\right)$ will keep symmetric during optimization. We can get the element-wise convergence relationship*

$$
\left|L_{i(k,l)}(t) - L_{i(k,l)}^*\right| \leq \begin{cases}
\left(m_i + 2k_i\right)\cdot\exp(-D\cdot t), & (k,l)\in\mathbb{C}_1\\
exp\left(-D\cdot t\right), & (k,l)\in\mathbb{C}_2\\
\left(m_i-1\right)\cdot\left(m_i+2k_i\right)\exp(-D\cdot t), & k=l
\end{cases}
$$

*where $\mathbb{C}_1 = \left\{(k,l)\mid k\neq l, \mathcal{T}_i\left(X\right)_{:,k}\neq\mathcal{T}_i\left(X\right)_{:,l}\right\}$, $\mathbb{C}_2 = \left\{(k,l)\mid k\neq l, \mathcal{T}_i\left(X\right)_{:,k}=\mathcal{T}_i\left(X\right)_{:,l}\right\}$,*

$$
L_{i(k,l)}^* = \begin{cases}
0, & (k,l)\in\mathbb{C}_1\\
\gamma, & (k,l)\in\mathbb{C}_2\\
-\sum_{l'=1,l'\neq l}^{m_i}L_{i(k,l')}^*, & k=l
\end{cases}
$$

*$L_{i(k,l)}(t)$ is the element of $L_i(t)$ at the $k$-th row and $l$-th column, $\mathbf{1}_{m_i\times m_i}$ is all one elements matrix. $\gamma = \frac{2}{\left|\{C_{\tilde{k},\tilde{l}}=0\}\right|} = \frac{2}{m_i+2k_i}$, $D$ is a constant defined in A.4 which equals to zero if and only if $X = I_{m_i\times n_i}$.*

*Proof.* We rewritten the gradient in proposition 2 with element wise formulation:

$$
\dot{W}_{\mathcal{T}_{i(k,l)}}\left(t\right) = \left(2Ca(t) - 4C_{k,l}\right)\cdot A_{i(k,l)}'(t),
$$

where $Ca(t) = \text{tr}\left(CA_i'\right)$ and the sub index denote the element in matrix.

With the assumption that $\left\|\mathcal{T}_i\left(X\right)_{k,:}\right\|_F^2 = 1$ and $\mathcal{T}_i\left(X\right)_{k,l} > 0$, we have $0 \leq \left(\mathcal{T}_i\left(X\right)\mathcal{T}_i\left(X\right)^\top\right)_{k,l} \leq 1$. Therefore $C_{k,l} = \left(\mathcal{T}_i\left(X\right)\mathcal{T}_i\left(X\right)^\top\right)_{k,k} - \left(\mathcal{T}_i\left(X\right)\mathcal{T}_i\left(X\right)^\top\right)_{k,l} \geq 0$ and specially $C_{k,k} = 0$, as $A_{i(k,l)}' = \frac{\exp(W_{ik,l})}{\|\exp(W_i)\|_1} > 0$, we have

$$
Ca(t) = \text{tr}\left(CA_i'\right) = \sum_{k=1,l=1}^{m_i} C_{k,l}A_{i(k,l)}'(t) > 0
$$

Denote $C_{\hat{k},\hat{l}} \in \min\limits_{k,l} C_{k,l}$, we have $C_{\hat{k},\hat{l}} \leq C_{k,l}$ and then consider

$$
\dot{W}_{\mathcal{T}_{i(\hat{k},\hat{l})}}(t) - \dot{W}_{\mathcal{T}_{i(k,l)}}(t)
$$
$$
= 2Ca(t)\left(A'_{i(\hat{k},\hat{l})}(t) - A'_{i(k,l)}(t)\right) - 4\left(C_{\hat{k},\hat{l}}A'_{i(\hat{k},\hat{l})}(t) - C_{k,l}A'_{i(k,l)}(t)\right)
$$

As we initialize $W_i(0) = \varepsilon \mathbf{1}_{m_i \times m_i}$, therefore $A'_{i(k,l)}(0) = \frac{1}{m_i^2}, \forall k,l$. Therefore $\dot{W}_{\mathcal{T}_{i(\hat{k},\hat{l})}}(0) - \dot{W}_{\mathcal{T}_{i(k,l)}}(0) = -4\left(C_{\hat{k},\hat{l}} - C_{k,l}\right)A'_{i(k,l)}(0) = -\frac{4}{m_i^2}\left(C_{\hat{k},\hat{l}} - C_{k,l}\right) \geq 0$. Then we have $W_{\mathcal{T}_{i(\hat{k},\hat{l})}}(t) \geq W_{\mathcal{T}_{i(k,l)}}(t)$ and $A'_{i(\hat{k},\hat{l})}(t) \geq A'_{i(k,l)}(t)$, the equal is token if and only if $t = 0$ or $C_{\hat{k},\hat{l}} = C_{k,l}$. Furthermore, $\dot{W}_{\mathcal{T}_{i(\hat{k},\hat{l})}}(t) - \dot{W}_{\mathcal{T}_{i(k,l)}}(t) \geq -4\left(C_{\hat{k},\hat{l}} - C_{k,l}\right)A'_{i(k,l)}(0)$, then $W_{\mathcal{T}_{i(\hat{k},\hat{l})}}(t) - W_{\mathcal{T}_{i(k,l)}}(t) \geq D_{\hat{k},\hat{l},k,l} \cdot t$, where $D_{\hat{k},\hat{l},k,l} = -4\left(C_{\hat{k},\hat{l}} - C_{k,l}\right)A'_{i(k,l)}(0) \geq 0$. Next, we consider

$$
A'_{i(\hat{k},\hat{l})}(t) = \frac{\exp(W_{i(\hat{k},\hat{l})})}{\left\|\exp(W_{\mathcal{T}_{i(k,l)}})\right\|_1}
$$
$$
= \frac{\exp(W_{i(\hat{k},\hat{l})})}{\sum\limits_{k,l}\exp(W_{\mathcal{T}_{i(k,l)}})} = \frac{1}{\sum\limits_{k,l}\exp(W_{\mathcal{T}_{i(k,l)}} - W_{\mathcal{T}_{i(\hat{k},\hat{l})}})}
$$
$$
\geq \frac{1}{\sum\limits_{k,l}exp\left(-D_{\hat{k},\hat{l},k,l}\cdot t\right)}
$$

As $C_{k,k} = 0$ and $C_{k,l} \geq 0$, therefore $C_{k,k} \in \min\limits_{k,l} C_{k,l} = 0$. It is not difficult to show that $C_{\hat{k},\hat{l}} = 0$ if and only if $\mathcal{T}_i(X)_{:,\hat{k}} = \mathcal{T}_i(X)_{:,\hat{l}}$. If $\left|\left\{C_{\hat{k},\hat{l}} = 0\right\}\right| = m_i + 2k_i$, then when $C_{\hat{k},\hat{l}} = 0$, $A'_{i(\hat{k},\hat{l})}(t) \geq \frac{1}{m_i + 2k_i + E_{\hat{k},\hat{l}}(t)}$, where $E_{\hat{k},\hat{l}}(+\infty) = 0$. Notice that $\sum\limits_{k,l} A'_{i(k,l)}(t) = 1$, we have

$$
\frac{1}{m_i + 2k_i + E_{\hat{k},\hat{l}}(t)} \leq A'_{i(\hat{k},\hat{l})}(t) \leq \frac{1}{m_i + 2k_i}
$$

Therefore, $A'_{i(\hat{k},\hat{l})}(+\infty) = \begin{cases} 0 & ,\mathcal{T}_i(X)_{:,k} \neq \mathcal{T}_i(X)_{:,l} \\ \frac{1}{m_i + 2k_i} & ,\mathcal{T}_i(X)_{:,k} = \mathcal{T}_i(X)_{:,l} \end{cases}$. Furthermore, $A_{i(\hat{k},\hat{l})} = 2A'_{i(\hat{k},\hat{l})} = \frac{2}{m_i + 2k_i} = \gamma$, $A_{i(k,l)}(+\infty) = \begin{cases} 0 & ,\mathcal{T}_i(X)_{:,k} \neq \mathcal{T}_i(X)_{:,l} \\ \gamma & ,\mathcal{T}_i(X)_{:,k} = \mathcal{T}_i(X)_{:,l} \end{cases}$. According to the definition of $L_i$,

$$
L^*_{i(k,l)} = L_{i(k,l)}(t)(+\infty) = \begin{cases} 0 & k \neq l, \mathcal{T}_i(X)_{:,k} \neq \mathcal{T}_i(X)_{:,l} \\ \gamma & k \neq l, \mathcal{T}_i(X)_{:,k} = \mathcal{T}_i(X)_{:,l} \\ -\sum_{l'=1,l'\neq k}^{m_i} L^*_{i(k,l')} & k = l \end{cases}
$$

Until now, we have prove that the adaptive regularization part of AIR-Net will convergence at the end. That is the upper bound of $\left|L^*_{i(k,l)} - L_{i(k,l)}(t)\right|$. Next we will focus on the convergence rate of AIR-Net. We discuss the rate under the three cases in the aforementioned formulation separately.

We simplify the notation furthermore before continue. $\mathbb{C}_1 = \left\{(k,l) \mid k \neq l, \mathcal{T}_i(X)_{:,k} \neq \mathcal{T}_i(X)_{:,l}\right\}$, $\mathbb{C}_2 = \left\{(k,l) \mid k \neq l, \mathcal{T}_i(X)_{:,k} = \mathcal{T}_i(X)_{:,l}\right\}$. Then the formulation is simplified as

$$
L^*_{i(k,l)} = \begin{cases} 0 & (k,l) \in \mathbb{C}_1 \\ \gamma & (k,l) \in \mathbb{C}_2 \\ -\sum_{l'=1,l'\neq k}^{m_i} L^*_{i(k,l')} & k = l \end{cases}
$$

If $(k,l) \in \mathbb{C}_2$ or $k = l$, we denote $D = \min D_{k,l}$, according to the definition of $\boldsymbol{E}_{k,l}(t)$, we have $\boldsymbol{E}_{k,l}(t) \le \exp(-D \cdot t)$

$$
\begin{aligned}
\left| \boldsymbol{A}^*_{i(k,l)} - \boldsymbol{A}_{i(k,l)}(t) \right| &= 2 \left[ \frac{1}{m_i + 2k_i} - \frac{1}{m_i + 2k_i + \boldsymbol{E}_{k,l}(t)} \right] \\
&\le 2 \frac{\boldsymbol{E}_{k,l}(t)}{\left(m_i + 2k_i\right)^2} \\
&\le 2 \cdot \frac{\frac{m_i(m_i-1)}{2} \cdot \exp(-D \cdot t)}{\left(m_i + 2k_i\right)^2} \\
&\le exp\left(-D \cdot t\right)
\end{aligned}
$$

Specially, when $(k,l) \in \mathbb{C}_2$ we have

$$
\left| \boldsymbol{L}^*_{i(k,l)} - \boldsymbol{L}_{i(k,l)}(t) \right| = \left| \boldsymbol{A}^*_{i(k,l)} - \boldsymbol{A}_{i(k,l)}(t) \right| \le exp\left(-D \cdot t\right)
$$

If $(k,l) \in \mathbb{C}_1$,

$$
\begin{aligned}
\left| \boldsymbol{L}^*_{i(k,l)} - \boldsymbol{L}_{i(k,l)}(t) \right| &= \left| \boldsymbol{A}^*_{i(k,l)} - \boldsymbol{A}_{i(k,l)}(t) \right| = \left| \boldsymbol{A}_{i(k,l)}(t) \right| \\
&\le \left| \sum_{(k',l') \in \mathbb{C}_1 \cup s C_3} \boldsymbol{A}_{i(k',l')}(t) \right| \\
&= \left| 2 - \sum_{(k',l') \in \mathbb{C}_2 \cup \mathbb{C}_3} \boldsymbol{A}_{i(k',l')}(t) \right| \\
&= \left| \gamma \cdot (m_i + 2k_i) - \sum_{(k',l') \in \mathbb{C}_2 \cup \mathbb{C}_3} \boldsymbol{A}_{i(k',l')}(t) \right| \\
&= \left| \sum_{(k',l') \in \mathbb{C}_2 \cup \mathbb{C}_3} \left( \gamma - \boldsymbol{A}_{i(k',l')}(t) \right) \right| \\
&\le \sum_{(k',l') \in \mathbb{C}_2 \cup \mathbb{C}_3} \left| \gamma - \boldsymbol{A}_{i(k',l')}(t) \right| \\
&= \sum_{(k',l') \in \mathbb{C}_2 \cup \mathbb{C}_3} \left| \boldsymbol{A}^*_{i(k',l')} - \boldsymbol{A}_{i(k',l')}(t) \right| \le (m_i + 2k_i) \cdot \exp(-D \cdot t)
\end{aligned}
$$

If $k = l$,

$$
\left| \boldsymbol{L}^*_{i(k,l)} - \boldsymbol{L}_{i(k,l)}(t) \right| = \left| \sum_{l'=1, l' \ne k}^{m_i} \left( \boldsymbol{L}^*_{i(k,l')} - \boldsymbol{L}_{i(k,l')}(t) \right) \right| \le (m_i - 1) \cdot (m_i + 2k_i) \cdot \exp(-D \cdot t)
$$

$\square$

### A.5 PROOF OF COROLLARY 1

**Corollary 1.** *In the setting of Theorem 2, we further have* $0 \le \mathcal{R}_{\boldsymbol{W}_i}(t) \le 2\left(m_i + 2k_i\right)\left(m_i - 1\right)m_i \cdot \exp(-Dt)$.

*Proof.*

$$
\begin{aligned}
\mathcal{R}_{\boldsymbol{W}_i}(t) &= \sum_{k,l} \boldsymbol{L}_{i(k,l)}(t)(t) \left\| \mathcal{T}_i(\boldsymbol{X})_{:,k} - \mathcal{T}_i(\boldsymbol{X})_{:,l} \right\|_F^2 \\
&\le 2 \cdot \sum_{k \ne l} \boldsymbol{L}_{i(k,l)}(t)(t) \le m_i \left(m_i - 1\right)\left(m_i + 2k_i\right) \cdot \exp(-Dt)
\end{aligned}
$$

$\square$

