# OpenReview forum: "AIR-Net: Adaptive and Implicit Regularization Neural Network for matrix completion"
_ICLR.cc/2022/Conference — ICLR 2022 Submitted_

### Official Review · Reviewer_MBKQ · 2021-11-02

**Correctness:** 4
**Technical Novelty And Significance:** 3
**Empirical Novelty And Significance:** 3
**Recommendation:** 5
**Confidence:** 4

**Main Review:**

##########################################################################
Summary:

This paper proposes a generalization and extension to deep matrix factorization as presented in the former paper from [Arora et al, NeurIPS 2019]. The extension allows more complex model which include inverse problems. The generalization part is build on a "vanishing" regularization which leads to better dynamics and convergence. These results are given by a theoretical analysis which highlight the effect of the proposed model. Finally the experiments illustrate the advantage of the model compared to state-of-art methods.


##########################################################################
Reasons for score:

Despite the interesting results in both theory and experiments, the presentation of the paper is too confused for publication (see cons below). Too many important details are in the appendix.

##########################################################################
Pros:

* The theoretical results are very interesting as they greatly improve the previous ones on deep matrix factorization.

* The Dirichlet Energy regularization term leading "vanishing" prior on the data is intriguing and a theoretically sound.

* The model is extensive to inverse problems and others. Important for application in image processing...

* The experiments shows clearly the advantage of the proposed model.

##########################################################################
Cons:

* Eq. (1) is an optimization problem not a model. The true model is hidden in the different part of the appendix (A.2 for the matrix factorization, A.3 for the regularization). Thus it is hard to understand (or even read) most of the paper since we have to switch between all the pages to catch the important pieces of the model.

* The theorems are very difficult to read as important information are in the appendix. Theorem 1 rely in on assumption which should be add to the text (I think there some place left). Theorem 2 is based on a regularization model which is only described in the A.3.


##########################################################################
Questions during rebuttal period:

Please address and clarify the cons above

#########################################################################
Some typos and others:

* Some sentences are confusing or need to be reshaped.
    - At the end of section 2.1, the details of the implicit low-rank are presented. It not a discussion.
    - In Theorem 2 at the end, I would rather write: "D is a constant which equals [...]".
    - End of page 4, "it's not difficult to find $R_{W_i^*}=0$" is confusing, I would rather write that "$R_{W_i^}(t)$ converge to 0".


**Summary Of The Paper:**

This paper proposes a generalization and extension to deep matrix factorization as presented in the former paper from [Arora et al, NeurIPS 2019]. The extension allows more complex model which include inverse problems. The generalization part is build on a "vanishing" regularization which leads to better dynamics and convergence. These results are given by a theoretical analysis which highlight the effect of the proposed model. Finally the experiments illustrate the advantage of the model compared to state-of-art methods.


**Summary Of The Review:**

This paper presents interesting results in both theory and experiments. Howeever the presentation of the paper is too confused for publication. Too many important details are in the appendix.

---

### Official Review · Reviewer_B6ib · 2021-11-02

**Correctness:** 3
**Technical Novelty And Significance:** 2
**Empirical Novelty And Significance:** 2
**Recommendation:** 5
**Confidence:** 3

**Main Review:**

**Quality:**

- Strengths: - The proposed method works fairly well empirically, beating other baselines using implicit regularization and also more traditional approaches.

- Weaknesses: - The paper can be quite hard to follow, see the clarity section below. -  Overall, the framework seems incremental. There does not seem to be a big distinction from [1], aside from parameterizing the dirichlet energy with learnable weights. The theoretical contributions appear minor and mirror those previously established in [2].

**Clarity:** The paper was hard to read and contained a number of typos. I have noted some here, but the manuscript contains others:
- page 1 abstract: “requires a regularization” -> “requires regularization”
- page 1: “over-parametric” -> “overparameterized”
- page 2: “The low-rank” -> “Low-rankness”
- page 2: “Furthermore, We” -> “Furthermore, we”
- page 3: The functionals $\mathcal{R}_{W_i}$ are missing the trace $\text{tr}( \cdot )$.
- page 3: “Another expected” -> “Another unexpected”
- page 5: “Specialy, we sigh two of $L_c(t=4000)$ out”
- page 6: “Air-Net Adaptive to Both Varies Data” should “varies” be “varying”?
- page 6 Figure 1: “the $t$-th row” -> “the $i$-th row”
- page 7: “reminding missing” -> “remaining missing”

**Novelty and significance:** The work seems to be an extension of [1], which also employed a deep matrix factorization approach to solve matrix completion problems and incorporated the dirichlet energy as an explicit regularizer. In that case, however, the laplacian was not learned, which seems to be the crucial difference. Some components of the theory is also borrowed from Arora et al [2].

**General comments:**

- At the top of page 3, it says that the structure of the network parameterizing the Laplacian matrix is discussed in A.4, but it appears to actually be in A.3. There is also no discussion on why the network is chosen in this particular fashion. It would be good to talk about why this network structure is used, e.g. the use of a softmax-type parameterization. Also, under this parameterization, is it even possible for the learned $L_i = 0$, the trivial solution?
- It feels misleading to say in Remark 1 that Theorem 2 requires no restriction on $\mathcal{T}_i$ or $X$. The first assumption is that all columns of $\mathcal{T}_i(X)$ are unit normed and the entries are positive. If, for example, $\mathcal{T}_i(X) := X$, then this means that for the result to hold, the columns of $X$ are unit normed and all of its entries are non-negative.
- The matrix $C_{k,l}$ should be defined in Theorem 2, since it arises in the definition of $\gamma$.
- I am a bit puzzled by Figure 1. This seems to indicate that as time progresses, the network learns that there should be less and less similarity between rows/columns, as indicated by the lack of dark regions as $t$ increases. What is the intuition for this? Shouldn't some notion of self-similarity be learned instead?

[1] Amit Boyarski, Sanketh Vedula, and Alex Bronstein. Spectral geometric matrix completion. MSML 2021

[2] Sanjeev Arora, Nadav Cohen, Wei Hu, and Yuping Luo. Implicit regularization in deep matrix factorization. NeurIPS 2019

**Summary Of The Paper:**

In this manuscript, the authors consider matrix completion problems. Leveraging recent advances on implicit regularization in (deep) matrix factorization problems, a new architecture for matrix completion is proposed. Specifically, the authors parameterize the unknown low-rank matrix as a deep linear network, which has been shown to exhibit a low-rank bias when learned via gradient descent. Additional regularization terms based on the Dirichlet energy are added to encourage other structural priors in the recovered solution, such as self-similarity between columns or blocks. The underlying Laplacian matrix is parameterized by learnable weights. The authors analyze the dynamics of gradient flow applied to their optimization problem and show empirically that this approach improves performance in certain settings.

**Summary Of The Review:**

Overall, while the proposed work shows empirical promise, it seems to be an incremental improvement over previous work. The readability of the manuscript can also be significantly improved. Based on these factors, my current score is a 5.

---

### Official Review · Reviewer_Untt · 2021-11-03

**Correctness:** 3
**Technical Novelty And Significance:** 3
**Empirical Novelty And Significance:** 2
**Recommendation:** 3
**Confidence:** 4

**Main Review:**

## Strengths:
1. Theoretically analyzing the gradient flow of the training loss to understand the properties of the proposed approach
2. good experimental results.

## Weaknesses
1. The paper is not easy to follow due to complicated notations. Also, it lacks a detailed description of some important concepts. For example, the end of the introduction mentions that $L$ in the adaptive regularizer is parameterized with DNN. However, I didn't find any detailed description for this except for the formulation $L_i(W_i)$ right after eq. (2). What is $L_i(W_i)$?

2. Without the detailed expression of $L_i(W_i)$  for the adaptive regularizer, the training loss in eq. (2) appears not well-posed. Particularly, the regularizers $trace(X L_r(W_r) X^\top)$ and $trace(X L_l(W_l) X^\top)$ are not bounded, i.e., they can go to negative infinity. Right after eq. (2), it says $R_{W_r}(X) = X L_r(W_r) X^\top$ which I think is not correct.

3. Theorem 1 analyzes the gradient flow when the regularizers are fixed, while Theorem 2 analyzes the gradient flow when the matrix is fixed. However, in practice, the entire training loss is minimized simultaneously, but no theoretical analysis is provided for this case.  With the regularizers fixed, Theorem 1 is similar to the existing results on analyzing the dynamic flow of deep matrix factorization, e.g., [Arora et al. 2019]. Difference and novelty compared to the existing work should be discussed.

4. The two theorems lack sufficient discussion to help the readers understand the main results. For example, what is the role of $L_r$ and $L_c$ in Theorem 1? How do they affect the results? The set $C_2$ in Theorem 2 consists of identical rows or columns, which seems empty in practice. If this is the case, Theorem 2 implies that the adaptive regularizer $L_i$ becomes a diagonal matrix. What is the role of the regularizer in this case?

5. Theorem 2 requires the matrix $X$ has positive elements, which is a very strong assumption and may not hold in practice.

**Summary Of The Paper:**

This paper studies the matrix completion problem where the goal is to recover the matrix from partially observed elements.  The proposed approach involves parameterizing the unknown matrix by deep matrix factorization and adaptive regularizers that are parameterized with deep neural networks. The authors studied the Adaptive and Implicit Regularization of the proposed approach which is called AIR-Net. Experiments are provided to demonstrate the effect of the proposed approach.

**Summary Of The Review:**

Overall, based on the detailed description above, the main contribution of this paper seems limited as the gradient flow of deep matrix factorization has already been studied. Also, the form of the adaptive regularize is not clear and the main results lack sufficient explanation.

---

### Official Review · Reviewer_Bmxi · 2021-11-03

**Correctness:** 4
**Technical Novelty And Significance:** 3
**Empirical Novelty And Significance:** 3
**Recommendation:** 5
**Confidence:** 3

**Main Review:**

- A general framework is proposed for studing matrix completion problems with regularization
- A good specific choice of implicit regularization based on the Dirichlet Energy with neural networks is proposed
- Theoretical analysis of the proposed algorithm indicates the expected system dynamcis
- Experimental evidence shows improvement on a variety of matrix completion tasks

Pros:
The results are interesting and compelling; indeed, there is clear improvement in a variety of tasks, and the dynamics of the reconstruction match the expectation in theory. It is also useful to draw links to other works through the more general formulation. Effective heuristics for regularization parameters are chosen so that they do not need to be tuned. The DE regularization has an intuitive impact.

Cons:
The experiments are rather limited to a small set of test cases (three images only) and it is not clear if the results are consistent across a larger data set.

There are also details missing, for example, a better explanation of the construction of the Laplace matrix in the DE regularization, and why it was chosen. Also, it is not clear that the experiment using fixed L at different iterations is so useful. Could instead the ideal L be used calculated from the fully sampled matrix?

It is also not entirely clear what value is added by generalizing other methods based on a very generic optimization equation (that essentially can capture any optimization problem). In particular, the main focus of the paper seems to be in the specific choice to regularize the DE of both the rows and the columns. However, no comparison is made to a situation where only one or the other is used. It is also not clear if overfitting can become an issue with enough iterations.

As a general comment, the writing is not always clear and could be improved to help clarity. There are also small errors, for example I believe "t-th row" in Figure 1 caption should say "i-th row".

Based on the preliminary results, I believe the paper is marginally below the acceptance threshold.

The justifications for this decision is the limited set of experiments justifying the choices made. Specifically, the value of adding the row- and column- DE regularizers, and the interplay with deep matrix factorization. In addition, the justification and explanation of the construction of Lr and Lc.



**Summary Of The Paper:**

This paper studies the problem of matrix completion using neural networks and deep matrix factorization as implicit and explicit regularization. The paper proposes a general framework and studies a specific case, namely, when the regularization is a form of Dirichlet Energy on the rows and columns of the matrix, and the matrix is formed as a product of L matrices. The results indicate improved performance of matrix completion under a variety of corruption methods and for a number of data sets.


**Summary Of The Review:**

In summary, the paper proposes a nice combination of implicit and explicit regularization for matrix completion using a deep matrix factorization and penalty on the DE of the rows and columns. The results are compelling, but some of the choices made are not fully justified or analyzed.

---

### Decision · Program_Chairs · 2022-01-20

**Decision:**

Reject

**Comment:**

There is a consensus that the contribution is not strong enough to effectively
argue for an important novel lead which would justify publication at ICLR.

Authors have also not engaged with the reviewers.

For these rejections, this paper cannot be endorsed for publication at ICLR 2022.